

# Body fat percentage is a key factor in elevated plasma levels of caffeine and its metabolite in women

Przemysław Domaszewski[1], Mariusz Konieczny[2], Paweł Pakosz[2], Jakub Matuska[3], Anna Poliwoda[1], Elżbieta Skorupska[3] and Manel Santafe[4]

[1] University of Opole, Opole, Poland
[2] Opole University of Technology, Opole, Poland
[3] Poznan University of Medical Sciences, Poznan, Poland
[4] Rovira i Virgili University, Reus, Spain

Corresponding author
Przemysław Domaszewski,
przemyslaw.domaszewski@uni.
opole.pl

## ABSTRACT

**Background:** Caffeine (CAF) intake is measured in absolute daily amounts or doses per kilogram of body weight. However, both methods are not entirely appropriate. The activity of liver enzymes that metabolise caffeine may be reduced in obese individuals, and plasma caffeine concentrations may vary depending on the fat-to-free-fat mass ratio. This study investigates the relationship between, between body fat percentage and the plasma concentrations of caffeine and its metabolites in women.
**Methods:** This experimental study with a between-group comparison design included 38 women (age 25.5 ± 2.7 years; body weight 66.5 ± 15.3 kg). Body fat percentage was measured using bioimpedance analysis (mBCA 515 SECA analyzer), and participants were categorized as non-obese (≤30% body fat, $n = 14$) or obese (>30% body fat, $n = 24$). Both groups received a dose of 6 mg/kg body weight of caffeine. Blood samples were taken 60 min after caffeine ingestion, and the concentrations of caffeine and its metabolites in plasma were determined by high-pressure liquid chromatography. The Mann-Whitney U test assessed CAF, paraxanthine (PX), and theobromine (TB) concentrations, while Spearman's correlation analyzed variable relationships. General linear model (GLM) compared caffeine metabolite levels, and GPower determined the required sample size (44) for large effects.
**Results:** The results showed that the obese group had significantly higher caffeine (Mdn = 10.64 *vs.* Mdn = 2.32) and PX concentrations (Mdn = 1.73 *vs.* Mdn = 0.85) compared to the non-obese group, with significant differences ($p < 0.001$ and $p = 0.007$, respectively). However, no significant difference in TB concentration was observed ($p = 0.486$). A linear model revealed that group membership significantly influenced CAF concentration ($p < 0.001$), explaining 56.8% of its variance. PX and TB concentrations showed poor model fits, with minimal explanatory power from group, age, fat mass, and body mass index (BMI). Correlation analysis found strong associations between CAF concentration and fat mass (rho = 0.689).
**Conclusions:** Higher body fat percentage is associated with increased plasma caffeine and paraxanthine concentrations following a weight-based caffeine dose. These findings suggest that body fat percentage may be a more relevant factor than total

body weight in caffeine metabolism, with potential implications for personalized caffeine dosing guidelines.

# INTRODUCTION

Global caffeine (CAF) consumption continues to increase due to changing lifestyles, hectic schedules and the constant need for quick bursts of energy (*Ariffin et al., 2022*). The effect of caffeine is primarily based on the antagonisation of the adenosine A1 and A2 receptors and the stimulation of the sympathetic branch of the autonomic nervous system. Other mechanisms include the inhibition of phosphodiesterase, which leads to increased cyclic adenosine monophosphate and the suppression of gamma-aminobutyric acid (GABA) receptors (*Surma et al., 2020*; *Domaszewski et al., 2021*). Caffeine is absorbed in the small intestine and reaches a bioavailability of 99% within about 45 min. About 80% of caffeine is oxidised into 1,7-dimethylxanthine (paraxanthine) during phase I, whereas about 16% is converted to 1,3-dimethylxanthine (theophylline) and 3,7-dimethylxanthine (theobromine). The half-life of caffeine is about 4 h, although it may be shorter in smokers and longer at higher doses or in people with liver dysfunction (*Alsabri et al., 2017*; *Nehlig, 2018*). The lethal dose of caffeine is between 5 and 10 grammes per day (*Alsabri et al., 2017*). Caffeine crosses the blood-brain barrier and reaches a concentration in the brain that corresponds to about 80% of its plasma concentration (*Kaplan et al., 1990*; *Alsabri et al., 2017*).

It is generally recognised that caffeine dosing should take into account adenosine receptor adaptation, CYP1A2 polymorphism, gender and body mass, but surprisingly, many official caffeine recommendations do not take these factors into account. A more optimal caffeine dosage takes into account body weight, which is often used for caffeine supplementation in sports, and the recommendations generally fall into three categories: low (less than or equal to 3 milligrammes per kilogramme of body weight), moderate (5 to 6 milligrammes per kilogramme of body weight) and high (greater than or equal to 7 milligrammes per kilogramme of body weight) (*Pickering & Kiely, 2018*; *Domaszewski, 2025*).

The question of the optimal caffeine dosage appears to be particularly important in a group of overweight or obese people. Determining the optimal caffeine dose is critical as studies show that individuals with a higher relative fat mass have significantly higher blood caffeine concentrations, a significantly higher caffeine absorption rate, a lower elimination rate and a longer plasma half-life of caffeine when the dosage is based on body weight compared to leaner individuals (*Kamimori et al., 1987*; *Skinner et al., 2014*).

Caffeine dosing models based solely on total body mass, regardless of body composition, may result in excessively high plasma caffeine levels in obese participants, which have been directly linked to adverse effects (*Kamimori et al., 1987*; *Skinner et al., 2014*). This phenomenon is probably related to the reduced activity of liver enzymes (cytochrome

P450) responsible for caffeine metabolism in obese people. A higher body fat percentage may impair enzyme activity, leading to caffeine metabolism and excretion changes. It leads to caffeine remaining in the body for longer, which increases its effect and the risk of overdose and side effects (*Brill et al., 2012*).

Adipose tissue can also act as a reservoir for substances such as caffeine, slowing its release and altering its pharmacokinetics (*Brill et al., 2012*, *2014*). This gradual release may prolong the stimulant effects of caffeine and increase the likelihood of adverse reactions such as increased heart rate, nervousness, anxiety, or sleep disturbances. In addition, caffeine essentially distributes in the aqueous cellular compartments, and muscle tissue contains more water than adipose tissue (*Bonati et al., 1982*). Therefore, body composition may influence the occurrence of caffeine-induced adverse effects, and individuals with a higher body fat percentage may have higher plasma levels of caffeine and its metabolites and process caffeine more slowly than individuals with a lower body fat percentage (*Massey, 1998*).

From a medical perspective, caffeine is the most widely consumed, unregulated and legally accessible psychoactive substance in the world and is considered safe and without significant adverse side effects (*Nehlig, Daval & Debry, 1992*). However, growing evidence suggests that optimising caffeine dosing based on body composition could maximise the benefits of caffeine while minimising the adverse effects (*Fisone, Borgkvist & Usiello, 2004*). Even relatively modest amounts of caffeine, when calculated concerning body weight, can cause a range of harmful and dangerous caffeine-related side effects in individuals with a higher proportion of fat mass compared to lean body mass.

The authors hypothesised that non-obese women would have lower plasma concentrations of caffeine and its major metabolites after ingestion compared to obese women when the caffeine dose is converted to mg/kg of total body weight. Confirmation of these hypotheses in future studies could pave the way for more precise caffeine dosing strategies, leading to less individual variability and greater consistency in achieving optimal ergogenic effects.

# MATERIALS AND METHODS

## Study participants

Forty-four women initially registered for the study. Six did not fulfil the inclusion criteria (two smokers, three women with caffeine hypersensitivity and one woman taking medication that may interact with caffeine). Thirty-eight adult women aged 22 to 31 years were included in the study. The subjects were categorised into two groups based on the criteria presented by *Okorodudu et al. (2010)*: non-obese ($n = 14$) with ≤30% body fat and obese ($n = 24$) with >30% body fat. The CONSORT flow diagram is shown in Fig. 1. The characteristics of the two groups are listed in Table 1. Additionally, 60% of participants reported using oral contraceptives, with 21% in the non-obese group and 83% in the obese group.

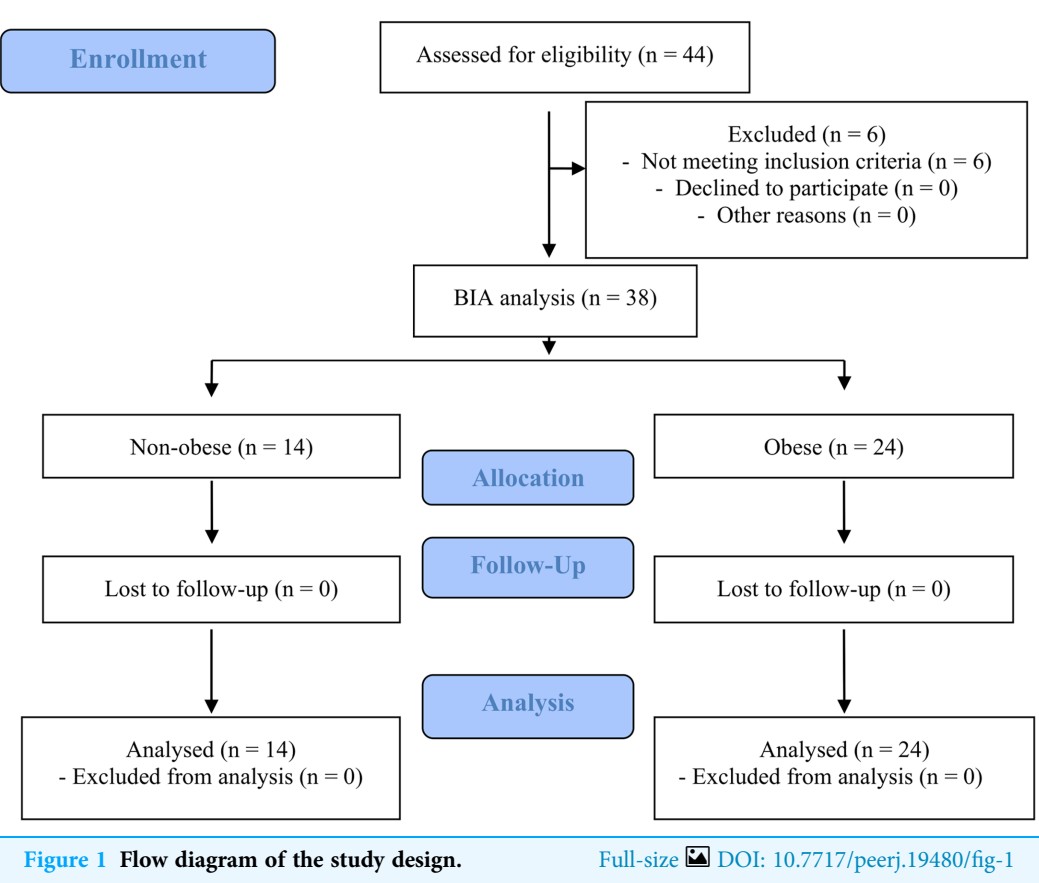

**Figure 1 Flow diagram of the study design.**

**Table 1 Participants characteristics.**

|  | All ($n$ = 38) (X ± SD) | Non-obese ($n$ = 14) (X ± SD) | Obese ($n$ = 24) (X ± SD) | p Non-obese *vs.* obese |
|---|---|---|---|---|
| Age (years) | 25.1 ± 2.7 | 23.9 ± 1.8 | 25.8 ± 2.9 | 0.043 |
| Height (cm) | 166.6 ± 6.4 | 164.4 ± 5.5 | 168.2 ± 6.6 | 0.081 |
| Weight (kg) | 66.5 ± 15.3 | 54.9 ± 5.6 | 73.2 ± 15.2 | <0.001 |
| BMI (kg/m$^2$) | 23.8 ± 5.0 | 20.3 ± 1.4 | 25.9 ± 5.1 | <0.001 |
| Relative fat mass (%) | 30.3 ± 8.0 | 22.3 ± 3.3 | 35.0 ± 6.1 | <0.001 |
| Absolute fat mass (kg) | 21.2 ± 10.9 | 12.3 ± 2.3 | 26.4 ± 10.5 | <0.001 |
| Fat-free mass (kg) | 45.3 ± 5.7 | 42.6 ± 4.5 | 46.9 ± 5.8 | 0.03 |
| Extracellular water (l) | 14.6 ± 2.0 | 13.2 ± 1.5 | 15.4 ± 1.8 | 0.001 |
| Intracellular water (l) | 29.4 ± 2.5 | 18.0 ± 2.0 | 20.2 ± 2.5 | 0.015 |

**Note:**
X, mean value; SD, standard deviation value; *p, p* value Mann-Whitney U test.

## Inclusion criteria

The participants met the following inclusion criteria: women, age between 18 and 31 years, absence of medical contraindications, non-smoking status, no hypersensitivity to caffeine (assessed by tolerance and use of coffee, caffeine supplements or energy drinks), informed consent to participate in the study, commitment to comply with all study guidelines, no

dependence on electronic life support systems (such as pacemakers or active prostheses), no current use of medications that could interact with caffeine metabolism, adherence to dietary restrictions (avoidance of caffeine, alcohol and cruciferous vegetables) prior to the study, good general health with no chronic medical conditions that could affect caffeine metabolism or the results of the study, no pregnancy or breastfeeding.

## Exclusion criteria

Failure to meet any of the inclusion criteria mentioned above.

## Assessment of caffeine habituation

The authors designed a food frequency questionnaire tailored to caffeine-containing products commonly consumed in the country where the study was conducted. Caffeine intake was classified based on the following consumption frequencies: more than three times daily, two to three times daily, once daily, five to six times weekly, two to four times weekly, once weekly, three times monthly, rarely, or never. This list included foods and beverages with high to moderate caffeine content, such as coffee, energy drinks, green and black tea, cocoa products, caffeine supplements, and certain medications. According to *Goncalves & Amiot (2017)*, 29% of participants were identified as high caffeine consumers (≥351 ± 139 mg/day), with a higher prevalence in the non-obese group (42%) compared to the obese group (21%). Moderate caffeine consumers (143 ± 25 mg/day) represent 32% of participants, including 14% of the non-obese group and 42% of the obese group. Low caffeine consumers (58 ± 29 mg/day) accounted for 39% of participants, with similar distributions in both groups (43% in the non-obese and 38% in the obese group).

## Body composition

Body composition was determined using the mBCA 515 SECA analyser (Seca, Germany). The (bioelectrical impedance analysis) BIA were repeated three times, and the arithmetic mean was calculated to minimise the error. Although BIA is not the most accurate method for determining body composition, it is most commonly used to determine body composition because it is a non-invasive and low-risk method that provides rapid results (*Hanley, Abernethy & Greenblatt, 2010*).

## Experiment design

Both groups received 6 mg of caffeine per kilogram of body weight, administered in transparent cellulose capsules and with water. The participants were instructed to abstain from caffeine and substances affecting caffeine metabolism, such as medication, cruciferous vegetables and alcohol, 48 h before the experiment. As caffeine is usually consumed in the morning and early afternoon, all measurements were taken between 7:00 and 12:00, *i.e.*, at least 2 h after a light meal. The test subjects were also asked to refrain from consuming caffeine-containing products on the day of the experiment and for the following 24 h. Blood samples were collected in 1.3 mL plasma tubes and allowed to clot. Subsequently, the samples were centrifuged at $5,500 \times$ g and 4 °C for 10 min. The resulting plasma was carefully extracted and aliquoted into 0.6 mL storage tubes. Each plasma sample was then frozen at −80 °C until further analysis.

## Ethics

The experiment was approved by the Bioethics Committee of the Poznan University of Medical Sciences (108/22), registered in the Australian-New Zealand Clinical Trials Registry (Ref. 12622000823774), and conducted according to the guidelines for research involving human subjects described in the Declaration of Helsinki. All study participants signed an informed consent form. All tests were performed in the physiological laboratory of the Poznan University of Medical Sciences.

## Chemicals

CAF, theobromine (TB), theophylline (TP), paraxanthine (PX) and caffeine—d9 (internal standard, IS), acetonitrile LS-MS grade, methanol LS-MS grade, water LS-MS grade and formic acid LS-MS grade were purchased from Merck (Poznan, Poland).

## LC-MS/MS analysis

All analyses were performed using a liquid chromatography system coupled to a microOTOF-Q II$^{TM}$ model mass spectrometer (Bruker Daltonics, Germany). Chromatographic separation was performed using an Acquity UPLC® C18 column (2.1 mm × 50 mm, 1.7 μm particle size) (Waters Corporation, Ireland) heated to 30 °C at a 0.3 mL/min flow rate. Ultrapure water with formic acid (0.2%, v/v) (A) and methanol (B) was used as the mobile phase. The run was performed in an isocratic mode with the mobile phase A and B concentrations set to 60% and 40%, respectively. The total analysis time was 6 min. The instrumental parameters were set as follows: capillary voltage, 4.5 kV; nebuliser gas flow (N2), 1.2 bar; desolvation line temperature, 300 °C; drying gas flow (N2), 8 L/min; and collision-induced dissociation gas pressure (Ar), 230 kPa. The analyses were performed in multiple reaction monitoring (MRM) mode. The mass spectrometer monitored the transition of CAF (m/z 195.1–138.0, RT 4.5 min), CAF-d9 (m/z 195.1 → 138.0, RT 4.5 min), CAF-d9 (m/z 204,14 → 144,1, RT 4.4 min), PX (181.1 → 124.1, RT 3.6 min), TP (181.1 → 123.9, RT 3.6 min) and TB (181.01 → 138.0, RT 3.1 min). The data were extracted with DataAnalysis Version 4.0 SP 5 (Bruker Daltonics).

## Method validation

The CAF, PX, TB and TP stock solutions were prepared individually in a mixture of methanol and water (1:1) at 100 ug/ml and stored at 4 °C. The stock solution of IS with a concentration of 1 mg/ml was prepared in methanol. Calibration standards were prepared by spiking 50 μl of blank plasma with the corresponding volumes of the mixed working solutions to obtain six calibration points for each analyte. Two calibration curves were prepared for caffeine, one for the concentration range 0.1–10 μg/ml and one for 10–25 μg/ml. It was 0.1–7 μg/ml for PX, TB and TP. Selectivity was assessed by comparing chromatograms of blank serum to ensure that no interfering substances were present at the retention times for CAF, PX, TB and TP. Linearity was assessed by plotting the analyte/IS peak area ratios against the analyte/IS *vs.* the nominated concentration by using a weighted $(1/x2)$ least–squares linearity regression analytes. Precision was expressed as RSD. Frozen plasma was thawed, and an aliquot of 50 μl plasma was spiked with 200 μl of CAF-d9

internal standard solution. After mixing and centrifugation (13,000 rpm, 5 min), the supernatant was transferred to a sample vial for LC-MS analysis.

## Statistical analysis

The parameters of the caffeine variables were statistically analysed as follows:

- First, the Shapiro-Wilk test was performed to assess the normality of the data.
- In order to verify the homogeneity of the variances, the Levene's test was conducted.
- Then, the Mann-Whitney U test was used for non-parametric variables: CAF concentration, PX concentration, and TB concentration and participants characteristics values.
- The median (Mdn) and interquartile rank (IQR) were calculated for each variable, as well as the effect size.
- The correlation between the variables was analyzed using Spearman's rank correlation.
- The comparison of caffeine metabolite levels between groups was analyzed using a general linear model (GLM) with covariance adjustments.
- The size of the intervention group was analyzed using the GPower 3.1.9.2 program. The sample size of a minimum of 44 women in each of the two groups (obese *vs* non-obese) was enough to detect a large effect (with 80% power and 5% significance level, two-sided).

All tests were performed using the free and open software JAMOVI, version 2.4.14. (https://www.jamovi.org).

# RESULTS

## Estimation of statistical differences in obesity rates across analyzed variables using the Mann-Whitney U Test

The obese group (predictor variable) had significantly higher caffeine concentrations (outcome variable) with a median (Mdn = 10.64) compared to the non-obese group (Mdn = 2.32), resulting in statistically significant differences between the groups (U = 8, $p < 0.001$). The PX concentration (outcome variable) was also higher in the obese group (Mdn = 1.73) compared to the non-obese group (Mdn = 0.85), with statistically significant differences between the groups (U = 78.5, $p = 0.007$). The difference in TB concentration (outcome variable) was not statistically significant (U = 145, $p = 0.486$) between the two groups. A detailed description of the presented parameters is provided in Table 2.

## The comparison of caffeine metabolite levels between groups was analyzed using a GLM with covariance adjustments

A linear model was fitted using the ordinary least squares (OLS) method to examine the effect of group, age, fat mass, and BMI on the dependent variable, CAF concentration, PX concentration and TB concentration.

The model examining the effect of group, age, fat mass, and BMI on CAF concentration explained 56.8% of the variance in the dependent variable, with an adjusted $R^2$ of 0.512.

**Table 2 Estimation of statistical differences between obesity rates in analysed variables.**

| Outcome variable | Predictor variable | Mdn | IQR | U | p | Effect size |
|---|---|---|---|---|---|---|
| CAF concentration (μg/ml) | Nonobese | 2.32 | 2.27 | 8 | <0.001 | 0.7 |
| | Obese | 10.64 | 5.44 | | | |
| PX concentration (μg/ml) | Nonobese | 0.85 | 0.94 | 78.5 | 0 | 0.4 |
| | Obese | 1.73 | 1.47 | | | |
| TB concentration (μg/ml) | Nonobese | 0.32 | 0.48 | 145 | 0.49 | 0.1 |
| | Obese | 0.31 | 0.47 | | | |

Note:
CAF, caffeine; PX, paraxanthine; TB, theobromine. Classification is based on body fat percentage, with individuals categorised as non-obese if their body fat percentage is less than 30% and obese if their body fat percentage is 30% or more. Med, median; IQR, interquartile range; U, Mann-Whitney U test value; p, p < 0.05 value; Mann-Whitney U.

This suggests that while the model fits the data reasonably well, other factors might also influence CAF variability. Among the independent variables, group membership had a significant effect on CAF concentration ($p < 0.001$, $\eta^2_p = 0.366$), while age, fat mass, and BMI did not show any significant relationship with the dependent variable.

For PX concentration, the model explained only 17.9% of the variance ($R^2 = 0.179$), and the adjusted $R^2$ of 0.073 indicates a poor model fit after accounting for the number of predictors. The model was not statistically significant, suggesting that group, age, fat mass, and BMI do not significantly explain the variation in PX concentration.

The model for TB concentration explained just 6.1% of the variance ($R^2 = 0.061$), and the adjusted $R^2$ value of −0.060 indicates that the model fit is poor. The negative adjusted $R^2$ suggests that the inclusion of the predictors worsened the model fit compared to using a simple mean model. This implies that group, age, fat mass, and BMI do not explain much of the variation in TB concentration (Table 3).

### The correlation between the variables was analyzed using Spearman's rho

The conducted correlation analysis shows that CAF concentration is strongly associated with absolute fat mass (Spearman's rho = 0.689, $p < 0.001$) and BMI (Spearman's rho = 0.654, $p < 0.001$), with moderate correlations observed with PX concentration correlation (Spearman's rho = 0.638, $p < 0.001$) and age (Spearman's rho = 0.361, $p = 0.030$). PX concentration is moderately correlated with TB concentration (Spearman's rho = 0.470, $p < 0.010$) and weakly with absolute fat mass (Spearman's rho = 0.340, $p < 0.05$) and BMI (Spearman's rho = 0.340, $p < 0.05$), but does not show a significant correlation with the age of the subjects. TB concentration does not show statistically significant relationships with other variables, suggesting that it may not be dependent on the factors examined in this analysis (Table 4).

## DISCUSSION

To the best of our knowledge, this study represents the first investigation of how obesity, considering body fat percentage, affects plasma concentrations of caffeine, theobromine, paraxanthine and theophylline 60 min after ingestion of 6 milligrammes of caffeine per

**Table 3 Results of the analyses of the variables CAF concentration, PX concentration, and TB concentration: ANOVA Omnibus tests for GLM models.**

| Dependent variable | Source | SS | df | F | p-value | η²p |
|---|---|---|---|---|---|---|
| **CAF concentration** | Model | 592.131 | 4 | 10.187 | 0.00002 | 0.568 |
| | Group | 260.339 | 1 | 17.916 | 0.00019 | 0.366 |
| | Age (years) | 8.805 | 1 | 0.606 | 0.44223 | 0.019 |
| | Absolute fat mass (kg) | 0.063 | 1 | 0.004 | 0.94778 | 0 |
| | BMI (kg/m²) | 0.003 | 1 | 0 | 0.98838 | 0 |
| **PX concentration** | Model | 8.256 | 4 | 1.69 | 0.17746 | 0.179 |
| | Group | 2.726 | 1 | 2.232 | 0.14531 | 0.067 |
| | Age (years) | 0.154 | 1 | 0.126 | 0.72495 | 0.004 |
| | Absolute fat mass (kg) | 0.025 | 1 | 0.021 | 0.88619 | 0.001 |
| | BMI (kg/m²) | 0.005 | 1 | 0.004 | 0.94865 | 0 |
| **TB concentration** | Model | 0.441 | 4 | 0.505 | 0.7324 | 0.061 |
| | Group | 0.273 | 1 | 1.249 | 0.27224 | 0.039 |
| | Age (years) | 0.1 | 1 | 0.459 | 0.5029 | 0.015 |
| | Absolute fat mass (kg) | 0.088 | 1 | 0.404 | 0.52977 | 0.013 |
| | BMI (kg/m²) | 0.003 | 1 | 0.016 | 0.90155 | 0.001 |

**Note:**

CAF concentration, PX concentration, and TB concentration, including the sum of squares (SS), degrees of freedom (df), F-values, p-values, and effect sizes (η²p).

kilogramme of body weight. Considering the dose and timing of blood sampling, it has been taken into account that most studies investigate the effects of caffeine approximately 60 min after ingestion, using a dose of 3 to 6 mg of caffeine per kg of body weight and remembering that plasma concentrations of caffeine in humans reach the highest blood levels approximately 1 h after caffeine administration (*Graham, 2001*; *Skinner et al., 2014*; *Kim, 2019*). However, in 1983, Blanchard and Sawers suggested that the highest plasma concentrations of caffeine may occur even earlier in healthy adult males (*Blanchard & Sawers, 1983*). It is difficult to compare these results with the existing literature as only a few previous studies investigate caffeine metabolism concerning obesity as measured by body fat percentage.

These results indicate that 60 min after the acute ingestion of caffeine at a dose of 6 mg per kilogramme of body weight, a statistically significant difference in the plasma concentrations of caffeine and its metabolites between non-obese and obese women was observed. In particular, the obese group showed significantly higher caffeine concentrations than the non-obese group. Paraxanthine concentrations were also significantly higher in the obese group than in the non-obese group. There were almost no differences in theobromine concentrations between the groups. Theophylline, another caffeine metabolite, was not detected in either group.

The relationship between obesity, determined by body fat percentage, and caffeine metabolism is intricate. Numerous studies indicate that caffeine metabolism is influenced by factors such as adenosine receptors, variations in the CYP1A2 gene and other gene polymorphisms (*i.e.*, ADORA2A, diet, gender and body mass) (*Kamimori et al., 1987*;

**Table 4  Correlation matrix.**

| | | CAF concentration | PX concentration | TB concentration | Absolute fat mass | BMI | Age |
|---|---|---|---|---|---|---|---|
| CAF concentration (μg/ml) | Spearman's rho | — | | | | | |
| | df | — | | | | | |
| | p-value | — | | | | | |
| PX concentration (μg/ml) | Spearman's rho | 0.638 | — | | | | |
| | df | 34 | — | | | | |
| | p-value | 0.00003 | — | | | | |
| TB concentration (μg/ml) | Spearman's rho | 0.324 | 0.470 | — | | | |
| | df | 34 | 34 | — | | | |
| | p-value | 0.05406 | 0.00382 | — | | | |
| Absolute fat mass (kg) | Spearman's rho | 0.689 | 0.340 | −0.073 | — | | |
| | df | 34 | 34 | 34 | — | | |
| | p-value | <0.00001 | 0.04258 | 0.67049 | — | | |
| BMI (kg/m²) | Spearman's rho | 0.654 | 0.340 | −0.011 | 0.917 | — | |
| | df | 34 | 34 | 34 | 34 | — | |
| | p-value | 0.00001 | 0.04266 | 0.95092 | <0.00001 | — | |
| Age (years) | Spearman's rho | 0.361 | 0.134 | −0.047 | 0.543 | 0.513 | — |
| | df | 34 | 34 | 34 | 34 | 34 | — |
| | p-value | 0.03068 | 0.43608 | 0.78716 | 0.00062 | 0.00138 | — |

*Brill et al., 2012*; *Skinner et al., 2014*; *Surma et al., 2020*; *Nadeem et al., 2021*; *Bulczak & Chmurzyńska, 2023*). However, only a few suggest that body fat percentage may determine plasma caffeine concentrations (*Kamimori et al., 1987*; *Brill et al., 2012*; *Skinner et al., 2014*). For this reason, we focused on this particular issue in our research, and the current study supports these findings and shows that the plasma caffeine concentration and probably also the occurrence of positive and potentially harmful effects of caffeine are influenced not only by generally recognised factors but also by body fat percentage. It should be pointed out that our results contradict various findings that caffeine metabolism is independent of obesity (*Nehlig, 2018*; *Pickering & Kiely, 2018*). The inconsistent results of previous studies on caffeine metabolism are likely due to differences in the methodological definition of obesity. In our studies, we used the criteria proposed by the American Society of Bariatric Physicians (*Okorodudu et al., 2010*), whereas the other studies relied primarily on BMI rather than directly measuring adipose tissue. The outdated BMI formula, developed by Quetelet nearly 200 years ago, is not an accurate measure of obesity but merely an inaccurate mathematical estimate (*Shah & Braverman, 2012*).

Given that high-fat mass, not high BMI is associated with chronic inflammation, acute phase responses and hepatic metabolic disorders, the results of previous studies should be re-evaluated. It is essential to recognise that reducing fat mass—and not just BMI—is directly associated with a lower metabolic and cardiovascular disease risk. In addition, increasing free fat mass offers numerous metabolic benefits (*Domaszewski et al., 2023*).

Nevertheless, an increased proportion of body fat with total body weight can potentially lead to an overdose of hydrophilic drugs if dosing is calculated based on total body weight (*Gouju & Legeay, 2023*). Reasonably, the dosage of caffeine, which has similar properties, should also be calculated considering body composition—fat mass and fat-free mass—and not total body weight. This perspective is at odds with reports suggesting that obesity has minimal effects on caffeine metabolism, meaning that the dosage of caffeine-containing drugs may not need to be significantly adjusted in individuals with obesity (*Nehlig, Daval & Debry, 1992*; *Nehlig, 2018*). Several mechanisms have been proposed to elucidate the observed differences in caffeine concentrations between obese and non-obese individuals. It is hypothesised that differences in individual body composition contribute to differences in the absorption rates (*Kamimori et al., 1987*). In addition, adipose tissue may serve as a reservoir for caffeine and other substances, leading to changes in plasma caffeine concentrations and a prolonged half-life of caffeine in obese individuals (*Brill et al., 2014*). Another factor is the reported impairment of cytochrome P450 enzyme activity in obese individuals, leading to slower metabolism and caffeine excretion (*Brill et al., 2012*). Combined with the relatively higher water content of lean individuals, which influences the distribution of caffeine in body fluids, this is further evidence of body composition's influential role on caffeine's pharmacokinetics.

Furthermore, the proposed mechanisms for ergogenic differences, adverse effects and differences in athletic performance between obese and non-obese individuals suggest that adenosine receptor activity is higher in obese than in lean individuals (*Skinner et al., 2010*), resulting in increased concentrations of caffeine and its metabolites in the bloodstream of obese participants (*Massey, 1998*). It is assumed that several other factors influence caffeine metabolism. Chronic caffeine consumption may lead to caffeine tolerance, although studies suggest that people generally develop a high tolerance (*Fredholm et al., 1999*; *Watson et al., 2016*). Individuals with different levels of habitual caffeine consumption do not show significantly different responses; however, non-consumers may be more sensitive to high doses of caffeine (*Graham, 2001*). Gender also may play a role: studies have shown that the effects of caffeine are different in men and women (*Domaszewski, 2023*). Although the enzyme CYP1A2, responsible for the breakdown of caffeine, is more active in men, studies have not shown consistent gender-specific differences in the plasma concentrations of caffeine and its metabolites (*Fredholm, 2011*). Age is another factor that has been considered, but current evidence suggests that age does not significantly affect caffeine metabolism (*Blanchard & Sawers, 1983*; *Massey, 1998*). Further studies are needed to fully elucidate the mechanisms and effects of these factors on caffeine metabolism.

In the study, significantly increased paraxanthine concentrations were also observed in the obese group compared to the non-obese group, emphasising its role as a primary metabolite of caffeine formed by CYP1A2 biotransformation (*Fredholm et al., 1999*). Like caffeine, paraxanthine effectively antagonises adenosine receptors and thus contributes to the pharmacological effects of caffeine (*Denaro et al., 1990*). Research suggests that paraxanthine concentrations usually reach the highest concentration approximately 8 to 10 h after caffeine ingestion (*Nehlig, 2018*). However, in our study, plasma levels of

paraxanthine in obese participants were significantly higher than in non-obese participants as early as 60 min after ingestion. The increased paraxanthine levels in obese are noteworthy because high paraxanthine plasma concentrations are associated with fewer adverse effects than high caffeine plasma concentrations (*Xing et al., 2021*; *Yoo et al., 2021*). In addition, unlike caffeine or its other metabolites, paraxanthine has been associated with potentiation of nitric oxide neurotransmission, which improves blood flow and may lead to positive changes in cardiovascular health (*Szlapinski et al., 2023*).

Our findings are of particular importance as caffeine is a common ingredient in fat-burning supplements, which overweight and obese women often consume. Therefore, it can lead to very high plasma concentrations of caffeine and its metabolites when administered per kilogramme of body weight. It should be noted that the magnitude of caffeine-induced side effects appears to correlate with plasma caffeine concentration (*de Mejia & Ramirez-Mares, 2014*), which can lead to severe interactions with the sympathetic nervous system and has been associated with gastrointestinal problems, headaches, tachycardia, anxiety, nervousness and insomnia in various groups—even with dangerous effects (*Wilk et al., 2019*; *Surma et al., 2020*; *Tiwari, 2020*; *Domaszewski, 2023*, *2025*). Conversely, athletes with a low body fat percentage may consume caffeine in doses too low to achieve the optimal ergogenic effects required to maximise athletic performance (*Domaszewski et al., 2021*; *de Souza et al., 2022*). Given the increasing global consumption of caffeine and the rise in obesity, understanding the effects of obesity on caffeine metabolism and the consequences could provide valuable insights for determining optimal caffeine dosing. Determining the minimum effective ergogenic dose in different populations is crucial but can vary considerably due to individual differences in caffeine metabolism, which are influenced by numerous factors. Identifying these factors could help minimise variability in caffeine response and improve personalised dosing strategies (*Skinner et al., 2014*). Future studies should include more extensive and detailed investigations considering CYP1A1 and ADORA gene polymorphisms for more accurate conclusions. Furthermore, the implications of these findings go beyond individual caffeine consumption and affect public health more broadly. Given the widespread use of caffeine in dietary supplements and medications, particularly in obese individuals seeking weight loss, the potential for varying metabolism according to body composition should be carefully considered. Excessive caffeine consumption in obese individuals, who may metabolise caffeine differently and have higher plasma concentrations, could lead to an increased risk of adverse effects. Therefore, healthcare providers should consider body fat percentage when recommending guidelines for caffeine intake to ensure that dosing is safe and effective for individuals with different body compositions. Public health campaigns should also aim to raise awareness of the potential risks of high caffeine consumption in obese individuals and advocate for a more personalised approach to the use of caffeine in diet and medicine.

In addition, these findings open up new avenues for research into the pharmacokinetics of other commonly used substances and drugs concerning body fat percentage. Similarly to the caffeine effect, the effect of plasma concentrations of other hydrophilic drugs may vary

depending on body composition. This finding could have significant implications for the dosing and efficacy of various drugs, particularly in obese individuals. As mentioned earlier, future studies should investigate these differences, focusing on especially various drugs to develop customised and precise dosing guidelines. By incorporating factors such as body fat percentage, gender differences, genetic polymorphisms and other individual characteristics, healthcare professionals can move towards more personalised medicine that optimises therapeutic outcomes and minimises adverse effects for each patient.

## CONCLUSIONS

In conclusion, obese women exhibited significantly higher concentrations of caffeine and paraxanthine when adjusted for body weight-dependent dosage, whereas theobromine levels remained unaffected by obesity. Moreover, age, fat mass, and BMI showed no significant influence on caffeine concentration. These findings suggest that caffeine metabolism may be more closely associated with body fat percentage rather than total body weight. Future large-scale studies should investigate whether caffeine consumption guidelines would be more effective if personalized based on an individual's body fat percentage rather than overall body weight.

### Funding
The authors received no funding for this work.

### Competing Interests
The authors declare that they have no competing interests.

### Author Contributions
- Przemysław Domaszewski conceived and designed the experiments, performed the experiments, analyzed the data, prepared figures and/or tables, authored or reviewed drafts of the article, and approved the final draft.
- Mariusz Konieczny performed the experiments, prepared figures and/or tables, and approved the final draft.
- Paweł Pakosz performed the experiments, prepared figures and/or tables, and approved the final draft.
- Jakub Matuska performed the experiments, prepared figures and/or tables, and approved the final draft.
- Anna Poliwoda performed the experiments, authored or reviewed drafts of the article, and approved the final draft.
- Elżbieta Skorupska performed the experiments, authored or reviewed drafts of the article, and approved the final draft.
- Manel Santafe analyzed the data, authored or reviewed drafts of the article, and approved the final draft.

## Human Ethics

The following information was supplied relating to ethical approvals (*i.e.*, approving body and any reference numbers):

The Bioethics Committee of the Medical University of Poznan (108/22).

## Ethics

The following information was supplied relating to ethical approvals (*i.e.*, approving body and any reference numbers):

The Bioethics Committee of the Poznan University of Medical Sciences (108/22).

## Clinical Trial Registration

This study was registered in the Australian-New Zealand Clinical Trials Registry (Ref. 12622000823774).

## Data Availability

The raw measurements are available in the Supplemental File.

## Supplemental Information

Supplemental information for this article can be found online at http://dx.doi.org/10.7717/peerj.19480#supplemental-information.

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
