# Peer review of "Body fat percentage is a key factor in elevated plasma levels of caffeine and its metabolite in women"

_PeerJ, doi:10.7717/peerj.19480_

## Round 0.1 · original submission · Major Revisions

· Academic Editor

Major Revisions

Dear authors,

Manuscript titled "Obesity as a key factor in elevated plasma levels of caffeine and its metabolite in women" that you submitted to PeerJ has been reviewed.
The reviewer(s) have suggested that some important points must be clarified and have requested substantial changes to be made in the manuscript. Therefore, I invite you to respond to the reviewer(s)' comments and revise your manuscript. The reviewer(s) comments are included at the end of this letter.

Please ensure that all review, editorial, and staff comments are addressed in a response letter and that any edits or clarifications mentioned in the letter are also inserted into the revised manuscript where appropriate.

Reviewer 1 ·

Basic reporting

(1) In the baseline table (Table 1), it is recommended to list the test statistics and P-values for the comparison between the two groups, and to supplement with comprehensive demographic information, such as caffeine consumption habits, use of oral contraceptives, and so on.
(2) The article emphasizes the importance of body fat percentage in caffeine metabolism, while obesity can be assessed through various aspects such as BMI, waist circumference, and body fat percentage. However, the author's viewpoint is not well highlighted in the title and abstract. It is suggested to make appropriate revisions to the title, abstract, and main body of the text. For example, the title could be revised to "Body fat percentage is a key factor in elevated plasma levels of caffeine and its metabolite in women" or something similar.
(3) In the discussion section, the author mentioned that this study was the first large-scale investigation, but in fact, the sample size included in this study was only 38 cases. Please pay attention to using appropriate terminology.

Experimental design

No comment.

Validity of the findings

(1) It is recommended to establish an additional grouping based on BMI, dividing individuals into obese (and/or overweight) and normal weight categories, and to compare whether there are differences in the concentrations of caffeine and its metabolites in the blood between the two groups. If no differences are found, this may add more credibility to the results.
(2) The analytical methods employed in the article are somewhat simplistic. It is recommended to enhance the credibility of the article by comprehensively utilizing multiple statistical analysis methods. For instance, further correlation analysis can be conducted to investigate the relationship between body fat percentage, BMI, and the concentrations of caffeine and its metabolites in the blood. Additionally, restricted cubic spline (RCS) analysis methods can be considered to explore the associations between body fat percentage, BMI, and caffeine and its metabolites.

Additional comments

None.

·

Basic reporting

Please confirm that all of the STROBE requirements are met; for example, for the first checklist, I could not find the study design clearly stated in both the title and abstract. I also could not find the sample size calculation. Please confirm and revise the manuscript based on the STROBE checklist and ensure all of the requirements are included.
The abstract could be improved by including key methodological details such as:
Study design (e.g., observational, cross-sectional, experimental).
Key variables (e.g., body fat percentage, caffeine intake, metabolite levels).
Instruments used (e.g., bioimpedance analysis for body fat, HPLC for caffeine analysis).
Statistical methods (e.g., correlation, regression, GLM).

Experimental design

If the aim is to assess the relationship between body fat percentage and caffeine metabolism (Lines 23-24), it would be more appropriate to treat body fat as a continuous variable rather than categorizing participants into two groups.

Correlation analysis (Pearson or Spearman) and multiple linear regression would provide a more accurate assessment of the relationship while allowing for confounder adjustments (e.g., caffeine intake, body weight).

If the goal is to compare caffeine metabolite levels between obese and non-obese individuals, a general linear model (GLM) with covariate adjustments would be a better alternative to Mann-Whitney U, which does not allow for confounder adjustments.

Validity of the findings

The Mann-Whitney U test is not the best choice for assessing relationships. A correlation analysis and regression model would be more appropriate.
Line 207 refers to this as a "large-scale study", but with n=38, this should be clarified. Consider either justifying the sample size with a power calculation or reframing the study as exploratory or pilot if appropriate.
The findings are scientifically relevant, but adjusting for confounding factors such as habitual caffeine intake, body weight, and age would improve the validity of the conclusions.

Additional comments

The study provides valuable insights, and with some refinements—particularly in statistical analysis, sample size justification, and reporting clarity—it will have a stronger impact.

---

## Round 0.2 · accepted · Accept

· Academic Editor

Accept

Dear Author,

Congratulations, after the good work of revisions in response to the reviewers' comments, I would like to inform you that your manuscript has been accepted for publication in PeerJ.